# Aptamer Efficacies for In Vitro and In Vivo Modulation of αC-Conotoxin PrXA Pharmacology

**DOI:** 10.3390/molecules24020229

**Published:** 2019-01-09

**Authors:** Germain Sotoing Taiwe, Jérôme Montnach, Sébastien Nicolas, Stéphan De Waard, Emmanuelle Fiore, Eric Peyrin, Tarek Mohamed Abd El-Aziz, Muriel Amar, Jordi Molgó, Michel Ronjat, Denis Servent, Corinne Ravelet, Michel De Waard

**Affiliations:** 1INSERM UMR 1087/CNRS UMR 6291, Institut du Thorax, Nouvelle Université à Nantes, LabEx Ion Channels, Science and Therapeutics, 8 Quai Moncousu, BP 70721 Nantes CEDEX 1, France; taiwe_sotoing@yahoo.fr (G.S.T.); jerome.montnach@univ-nantes.fr (J.M.); sebastien.nicolas@univ-nantes.fr (S.N.); stephan.dewaard@univ-nantes.fr (S.D.W.); michel.ronjat@univ-nantes.fr (M.R.); 2Department of Zoology and Animal Physiology, Faculty of Sciences, University of Buea, P.O. Box 63, Buea, Cameroon; 3CNRS, DPM UMR 5063, University Grenoble Alpes, 38041 Grenoble, France; emmanuelle.fiore@univ-grenoble-alpes.fr (E.F.); eric.peyrin@univ-grenoble-alpes.fr (E.P.); corinne.ravelet@univ-grenoble-alpes.fr (C.R.); 4Zoology Department, Faculty of Science, Minia University, 61519 El-Minia, Egypt; tarek.mohamed@mu.edu.eg; 5Service d’Ingénierie Moléculaire des Protéines, Institut des Sciences du Vivant Frédéric Joliot, Commissariat à l’Energie Atomique, Université Paris-Saclay, F-91191 Gif sur Yvette, France; muriel.amar@cea.fr (M.A.); jordi.molgo@cea.fr (J.M.); denis.servent@cea.fr (D.S.); 6Institut des Neurosciences Paris-Saclay, UMR 9197, CNRS/Université Paris-Sud, 91198 Gif sur Yvette, France; 7Smartox Biotechnology, 6 rue des Platanes, 38120 Saint Egrève, France

**Keywords:** αC-conotoxin PrXA, DNA aptamer, oligonucleotide, toxin neutralization, cone peptide, venom

## Abstract

The medical staff is often powerless to treat patients affected by drug abuse or misuse and poisoning. In the case of envenomation, the treatment of choice remains horse sera administration that poses a wealth of other medical conditions and threats. Previously, we have demonstrated that DNA-based aptamers represent powerful neutralizing tools for lethal animal toxins of venomous origin. Herein, we further pursued our investigations in order to understand whether all toxin-interacting aptamers possessed equivalent potencies to neutralize αC-conotoxin PrXA in vitro and in vivo. We confirmed the high lethality in mice produced by αC-conotoxin PrXA regardless of the mode of injection and further characterized myoclonus produced by the toxin. We used high-throughput patch-clamp technology to assess the effect of αC-conotoxin PrXA on ACh-mediated responses in TE671 cells, responses that are carried by muscle-type nicotinic receptors. We show that 2 out of 4 aptamers reduce the affinity of the toxin for its receptor, most likely by interfering with the pharmacophore. In vivo, more complex responses on myoclonus and mice lethality are observed depending on the type of aptamer and mode of administration (concomitant or differed). Concomitant administration always works better than differed administration indicating the stability of the complex in vivo. The most remarkable conclusion is that an aptamer that has no or a limited efficacy in vitro may nevertheless be functional in vivo probably owing to an impact on the biodistribution or pharmacokinetics of the toxin in vivo. Overall, the results highlight that a blind selection of aptamers against toxins leads to efficient neutralizing compounds in vivo regardless of the mode of action. This opens the door to the use of aptamer mixtures as substitutes to horse sera for the neutralization of life-threatening animal venoms, an important WHO concern in tropical areas.

## 1. Introduction

One of the main issues with: (i) drug abuse (accidental or deliberate), (ii) intoxication (often from alimentary sources), (iii) poisoning (criminal cases) or (iv) envenomation is to find the most appropriate medical response to treat patients that are often in critical and emergency conditions. Up to now, the most appropriate response, at least for the cases of envenomation, was the treatment of patients with immunoglobulin G injection purified from blood serum of venom-immunized animals [1]. While this technique works well, despite the fact that it was invented a century ago [2], it possesses many drawbacks: less and less companies are involved on this niche market, ever increasing prices for the therapies or difficulties in market access [3], mostly incompatible with several national health budgets, animal suffering during the immunization steps, limited shelf-life, lack of immune response for some toxins, risks of serum sickness or worse anaphylactic shocks because of the Fc fragment of animal origin. While progress is still expected to occur with the antivenom sera technology, such as the production of a serum effective against the combined venom of several venomous species, the implementation of Good Manufacturing Practices for their production, the inclusion of preservatives to avoid fungal and bacterial contamination, the removal of Fc fragments by pepsin or papain digestion, the freeze-drying of the antivenom to ensure its stability on the shelf, and the humanization of the antibodies and use of recombinant toxins [4], all these measures are expected to further raise the costs of the antivenom production (see the World Health Organization publication entitled “Guidelines for the production, control and regulation of snake antivenom immunoglobulins”) [5]. Concerning toxins of bacterial origin (for instance anthrax, botulism), significant efforts have been invested in producing effective antibodies against these toxins that are considered as potential bioweapons in several countries. The case of anthrax is best exemplifying these efforts of neutralizing antibody production. Clinical antibodies against anthrax include Anthrivig^®^ by Emergent (Rockville, MD, USA), Raxibacumab^®^ from Human Genome Services (also in Rockville, MD, USA), Valortim^®^ from PharmAthene Inc. (Annapolis, MD, USA), Anthim^®^ from Elusys Therapeutics Inc. (Pine Brook, NJ, USA) and Thravixa^®^ again from Emergent. All of these antibodies were the subject of clinical trials in 2011 [6]. These antibodies represent a faster alternative to vaccination programs that are also more difficult to justify considering the risk to benefit ratio vaccination implies. Another promising approach to neutralize life-threatening bacterial toxins is the clinical development of compounds that inhibit the retrograde transport from the early endosome to the trans-Golgi network without affecting cell viability or normal function [7].

More recently, we have emphasized an alternative methodology to antibody production, which relies on the use of DNA aptamers to neutralize toxins [8]. A first hint that aptamers could be efficient in neutralizing toxins in vitro was published in 2000 with the discovery of ricin-inhibiting RNA aptamers active on the catalytic A-chain [9]. For the purpose of our own demonstration, we have chosen to work with a selective α-conotoxin. This family of cysteine-rich peptides present in most of the venoms of the predatory marine cone snails (genus *Conus*) has been greatly characterized, and is known to target distinct isoforms of nicotinic acetylcholine receptors (nAChRs) found at the skeletal neuromuscular junction and in the peripheral and central nervous systems (for a review see reference [10]). The selective αC-conotoxin PrXA, was chosen because it was demonstrated as a highly selective and potent antagonist of the mouse muscle-type nAChR receptor [11]. The binding of this peptide is reversible which is probably a valuable quality when assessing toxin-neutralizing compounds. In addition, it competes with α-bungarotoxin binding most essentially at the α/δ subunit interface, providing another assay for the evaluation of aptamers. In our previous report, we had isolated several αC-conotoxin PrXA-binding DNA aptamers that were classified along three groups of homolog sequences [8]. While we demonstrated that one of these aptamers could successfully prevent αC-conotoxin PrXA binding onto muscle-type nicotinic receptors and in vitro and in vivo effects on muscle contraction and lethality, respectively, we had not investigated the effects of other aptamers. We now extended our observations to several other αC-conotoxin PrXA-binding aptamers by investigating their effects on new paradigms such as the ability of the toxin to block ACh-mediated currents in TE671 cells or toxin-induced abdominal myoclonus in mice, along with toxin-induced mice lethality. The data demonstrate that the toxin-binding aptamers can be classified according to two categories: (i) those that alter the toxin efficacy in vitro by negatively shifting the IC_50_ of toxin block of ACh-mediated currents and that are consequently active in vivo, and (ii) those that are fully inactive in vitro, presumably because they don’t bind on the pharmacophore of the peptide. The first category can further be subdivided into two types of aptamers: (i) those active only upon co-injection of both the toxin and the aptamer in vivo, and (ii) those that are active both upon co-injection or if the aptamer is injected as a therapeutic compound to reverse the toxic effects of the peptide.

## 2. Results

### 2.1. αC-Conotoxin PrXA Lethality and Earlier Data on Aptamer Neutralization

We previously reported that αC-conotoxin PrXA injection following intraperitoneal (i.p.) or subcutaneous (s.c.) injection induces fast killing in mice (less than 3 min regardless of the mode of injection) [8]. Here we observed that intravenous (i.v.) injection of αC-conotoxin PrXA dose-dependently induces killing of *Mus musculus* Swiss mouse (Appendix A) and that increasing concentrations of the toxin shorten the delay for death occurrence (Appendix A). αC-conotoxin PrXA-induced mouse lethality occurred with a LD_50_ of 0.011 ± 0.002 µg/g body weight (*n* = 6 for each concentration). Maximal lethality was observed after administration of 0.5 µg/g body weight. These results are quite comparable to those obtained with subcutaneous injection in the same mouse line [8]. The fastest death occurrence occurs within 1.17 ± 0.28 min following intravenous injection of 1.5 µg/g body weight. The toxin concentration required to produce a twofold decrease in death latency was 0.007 ± 0.015 µg/g body weight (*n* = 6 for each concentration). Compared to intraperitoneal or subcutaneous injections [8], intravenous injection decreases latency to death (up to 7 times faster) without diminishing the doses required to increase lethality. We conclude therefore that intravenous injection is more effective to look at αC-conotoxin PrXA toxicity and kept this experimental paradigm. As a reminder of earlier results, we also illustrate that the aptamer D7, previously selected for its interaction with αC-conotoxin PrXA [8], is able to prevent the αC-conotoxin PrXA-mediated inhibition of [^125^I]-α-bungarotoxin binding onto Torpedo muscle-type nicotinic receptors, indicating that it should bind onto and mask the pharmacophore of the toxin (Appendix A). In addition, the D7 aptamer efficiently reverses the αC-conotoxin PrXA-mediated inhibition of muscle contractions induced by nerve stimulation of the isolated mouse phrenic-hemidiaphragm nerve-muscle preparation (Appendix A). As we previously identified several αC-conotoxin PrXA-binding aptamers, we investigated in vitro and in vivo on new experimental paradigms their ability to modify the efficacy of the toxin.

### 2.2. High Throughput Evaluation of ACh and αC-Conotoxin PrXA Effects on the TE671 Cell Line

The TE671 human cell line is derived from rhabdomyosarcoma and naturally expresses muscle-type nicotinic acetylcholine receptors (nAChR) and desmin, an intermediate filament protein [12]. Further proof that these cells express the muscle-type nAChR comes from the fact that the biophysical, immunological and biochemical properties of the channel are muscle type, along with the fact that these cells bind α-bungarotoxin, a hallmark of muscle type AChR [13]. We further wanted to test the value of this cell line in investigating the nature of the nACh response and the activity of αC-conotoxin PrXA, a toxin that is reportedly highly active and selective of muscle-type nAChR [10]. For these evaluations we benefited from a high-throughput automated patch clamp (Syncropatch 384, Nanion, Munich, Germany) to record from hundreds of TE671 cells and get a statistical view of ACh-mediated currents. A first hint of the value of TE671 cells in patch recordings was provided by measuring the individual cell capacitance (Appendix A). The distribution of cell capacitance was Gaussian with a mean value of 15.9 ± 6.9 pF (SD, *n* = 1444 cells). As a starting point, we evaluated the ACh concentrations required to elicit a saturating response of the TE671 cells. The membrane potential was held at −60 mV and the K^+^ concentrations set as to minimize the ionic contribution of Ca^2+^-sensitive SK potassium channels occurring in response to nAChR activation. As shown, increasing concentrations of ACh induces gradually increasing inward currents in TE671 cells that possessed a clear tendency to desensitize at the highest concentrations of ACh and during the application time (Figure 1a). An evaluation of the histogram of current density distribution at maximal ACh concentration (100 μM) demonstrates the rather non-Gaussian distribution of TE671 cell responses, indicative of the degree of cell population heterogeneity (Figure 1b). 47.9% of the cells had current amplitudes lower than 100 pA (*n* = 1444 cells). Mean current density was 17.7 ± 0.9 pA/pF (at 100 μM ACh, SEM, *n* = 1444 cells). Fit by an exponential decrease of the data yielded a constant value of 17.2 pA/pF, corroborating the mean current density. The dose-response curve of ACh-mediated inward current amplitudes (*n* = 167 cells) indicate submaximal activation at 10 μM ACh and an EC_50_ of current activation of 3.1 ± 3.9 μM (Figure 1c). The current amplitude range in response to ACh, the desensitization and the dose-response curve are all in full agreement with earlier published data using manual patch clamp [14], thereby validating the automated patch clamp approach. The concentration of 100 μM ACh was chosen for all experiments afterwards to ensure maximal activation of the muscle-type nAChR. The effect of αC-conotoxin PrXA cannot be assessed directly on the ACh-mediated response because it desensitizes too fast to allow a kinetically compatible assessment of toxin-mediated ionic current block. To assess the blocking effects of the toxin, cells were thus first pre-incubated 20 min with variable concentrations of αC-conotoxin PrXA (to leave enough binding time, especially at low concentrations of the toxin) and the residual response to ACh was evaluated. In order to ensure that each cell was its own control, to avoid the intrinsic variability in current amplitude response we observed among cells (Figure 1b), we established a washing protocol aimed at removing the toxin, followed by a second application of ACh that mediates a response that was considered as the intrinsic control condition. This protocol takes advantage of the demonstrated reversibility of the interaction of αC-conotoxin PrXA with its binding site [11]. Use of this protocol in the absence of αC-conotoxin PrXA indicates that the second application of ACh mediates a response of equivalent amplitude but with a lower tendency to desensitize (see representative traces in Figure 1d). The average peak current densities, mediated by 100 μM ACh, were compared for the first and second applications. As shown, these densities were not significantly different with values of −56.7 ± 10.2 pA/pF (*n* = 25) for the first application *versus* −48.0 ± 8.6 pA/pF (*n* = 25) for the second application (marginal difference of 1.18-fold) (Figure 1e). In addition, all the data obtained for the αC-conotoxin PrXA applications were normalized to this control condition further limiting the sources of misevaluation. For all these analyses, currents lower than 100 pA at the second application of 100 μM ACh were discarded because they were a source of misevaluation of the amplitude of αC-conotoxin PrXA effects. As shown, the second application of ACh mediated a progressively larger response than the first application in conditions where αC-conotoxin PrXA concentration was increasing, indicating a greater removal from current block by the toxin upon its washout (Figure 1f, see arrows). Using this protocol, we were able to reconstruct a dose-response curve for the αC-conotoxin PrXA-mediated block of ACh response (Figure 1g). Fit of the experimental data yields an IC_50_ of current block 6.9 ± 1.5 nM and a Hill coefficient of 1.2 ± 0.4 (*n* = 80 cells). In addition, average current amplitudes in response to the second application of ACh were −519.0 ± 99.2 pA (after 100 nM αC-conotoxin PrXA, *n* = 11 cells) *versus* −510.9 ± 132.6 pA (after 0.3 nM αC-conotoxin PrXA, *n* = 11 cells) indicating excellent washing conditions of the toxin and reversibility of the channel block. Our findings are in excellent agreement with the previously reported IC_50_ values for adult and fetal muscle nAChR (1.8 and 3 nM, respectively) [11]. It is also coherent with the reported dose-dependence of αC-conotoxin PrXA-mediated inhibition of mouse muscle contraction (IC_50_ of 23 nM) [8]. In addition, they demonstrate for the first time the efficacy of the toxin on the human muscle-type nAChR.

### 2.3. Evaluation of Aptamer Efficacies in Neutralizing αC-Conotoxin PrXA-Mediated Inhibition of ACh Response

In an earlier investigation we had identified several DNA aptamers that recognize αC-conotoxin PrX and demonstrated the in vitro and in vivo efficacy of them [8]. Yet, these aptamers all possess different sequences (Figure 2a), and were classified within three families according to sequence homologies (D7, family 3; B4, family 2; and D3 and A5, family 1) [8]. Their secondary structures were previously predicted using the Mfold web server [15] and all shown to possess stem loop structures [16] rather than G-quadruplex structures. Because of sequence and structural differences between the aptamers, it is likely that their interaction with αC-conotoxin PrXA may differ, as well as their neutralizing efficacies. In particular, the differentiating factors between these aptamers are related to the following matters: (i) how well they mask the pharmacophore of the peptide and interfere with its binding capacity on muscle-type nicotinic receptors and (ii) how efficiently these aptamers may interact with the toxin in a complex environment in vivo and neutralize its toxic effects. In order to investigate the neutralizing capabilities of the aptamers, we first tested their effects in vitro using our validated patch clamp assay. As a control, we initially determined whether each of the aptamers would affect non-specifically the response of the muscle-type nAChR in TE671 cells. As shown, a 20 min incubation of TE671 cells with 300 nM aptamer did not preclude an ACh response for the first application of ACh and did not alter the relative amplitude of the second ACh response after thorough washing of the aptamers (Appendix A). The first (in the presence of each aptamer) and second (in its absence, after aptamer washout) applications of ACh thus yielded the same current amplitudes similarly to what was observed in the absence of aptamers (see Figure 1d,e). Quantification of these effects demonstrates the lack of a significant alteration of ACh responses by the aptamers (Appendix A). These results are expected since these aptamers were neither raised against ACh or the nAChR, and therefore are indicative of the lack of non-specific effects. Next, we investigated how a single concentration of the aptamers (300 nM) may affect the blocking responses of variable concentrations of αC-conotoxin PrXA. Both entities were pre-incubated for 30 min before being co-applied to the TE671 in the Syncropatch 384 recording unit. As shown in a representative example, a 82.8% inhibition of ACh response is observed in the presence of 30 nM αC-conotoxin PrXA and in the absence of aptamers, as judged by the response amplitude to the first application of ACh *versus* the second one (after extensive washout of αC-conotoxin PrXA) (Figure 2b; see also Figure 1f). In contrast, if the aptamers D7 or B4 were co-applied with αC-conotoxin PrXA, the amplitude of ACh current inhibition was lessened, suggesting that in these conditions they neutralized to some extent the effect of the toxin. This was not observed for the two other aptamers, A5 and D3, that both seemed to leave intact the blocking potential of the toxin (Figure 2b). The same experiments were repeated on a large number of cells and using variable concentrations of the toxin and the same fixed concentration of each aptamer. As shown, the main effect of aptamers D7 and B4 were to significantly shift the dose-response curve of αC-conotoxin PrXA for ACh response block (Figure 2c). Fit of the data reveal IC_50_ values of 30.9 ± 2.2 nM for αC-conotoxin PrXA with D7 (*n* = 108 cells) and 26.5 ± 1.7 nM for αC-conotoxin PrXA with B4 (*n* = 106 cells), representing a 4.5- or 3.9-fold reduction in affinity, respectively. In contrast, co-application of the A5 or the D3 aptamers had a minimal impact of the dose-response curve of αC-conotoxin PrXA block. IC_50_ values were not significantly different (5.8 ± 1.2 nM for αC-conotoxin PrXA with A5 (*n* = 46 cells) and 9.0 ± 1.5 nM for αC-conotoxin PrXA with D3 (*n* = 53 cells). Based on these data we conclude that the D7 and B4 aptamers have the ability to mask the pharmacophore of αC-conotoxin PrXA, at least partially, while the A5 and D3 aptamers do not. We next set to investigate the effects of these aptamers in vivo to see how these alterations (D7 and B4 aptamer-mediated) may translate in reduced toxicity.

### 2.4. Mice Intoxication Symptoms Induced by αC-Conotoxin PrXA

Behavioral changes of each animal were observed following intravenous injection of αC-conotoxin PrXA at the doses of 0.005, 0.008, 0.01, 0.05, 0.1, 0.5, 1 and 1.5 µg/g of body weight. Signs and symptoms which occurred in response to the αC-conotoxin PrXA were hypoactivity, salivation, tachypnea, tremors, loss of the righting reflex, myoclonus, exophthalmos, salivation and syncope, depending on toxin concentration (Table 1). Mice that died from a lethal dose of αC-conotoxin PrXA showed signs of respiratory failure (decreased respiratory rate and irregular breathing) before death. Organs of dead animals of each group did not show any unusual signs and were normal in both size and color.

### 2.5. Combined or Separated Injection of Aptamers Neutralize Toxin-Induced Abdominal Myoclonus

Behavioral observation of mice after intraperitoneal injection of αC-conotoxin PrXA indicated the occurrence of: (i) clonic contractions of abdominal muscle only; (ii) both tonic and clonic contractions of abdominal muscle; or (iii) tonic contractions of abdominal muscle only. We tested the ability of the aptamers to neutralize toxin-induced abdominal myoclonus when toxin and aptamers are injected in the same mixture or separately. Intraperitoneal injection of 0.5 µg/g body weight of αC-conotoxin PrXA induce a 120-fold increase in the number of abdominal myoclonus per unit of time compared to control. For both aptamers, when toxin and aptamers were injected in the same mixture, the number of abdominal myoclonus per unit of time was significantly and dose-dependently reduced compared to the injection of the toxin alone (aptamer B4 (8 µg/g body weight) in blue: 0.45 ± 0.02/min abdominal myoclonus *versus* 13.94 ± 0.61/min abdominal myoclonus for toxin only; aptamer D7 (8 µg/g body weight) in green: 0.13 ± 0.02/min abdominal myoclonus *versus* 12.83 ± 0.58/min abdominal myoclonus for toxin only; *n* = 6 for each concentration) (Figure 3a). When the toxin is injected first and the aptamer injection follows 1 min later in the mouse tail vein, the number of abdominal myoclonus per unit of time was also significantly reduced compared to the injection of the toxin alone (aptamer B4 (8 µg/g body weight) in blue: 1.04 ± 0.05/min abdominal myoclonus *versus* 13.44 ± 0.27/min abdominal myoclonus for toxin only; aptamer D7 (8 µg/g body weight) in green: 0.47 ± 0.04/min abdominal myoclonus *versus* 12.61 ± 0.36/min abdominal myoclonus for toxin only; *n* = 6 for each concentration) (Figure 3b). In contrast, the aptamers D3 and A5 did not produce any reduction of αC-conotoxin PrXA-induced myoclonus at the concentration of 8 μg/g of body weight (data not shown), which is coherent with the lack of effects of these two aptamers on the dose-response of toxin-mediated block of ACh currents in TE671 cells. We therefore concluded that the aptamers B4 and D7 efficiently neutralized αC-conotoxin PrXA effects on abdominal myoclonus when they were co-injected or injected separately in time.

### 2.6. Combined or Separated Injection of Aptamers Neutralize Toxin-Induced Lethality

The myoclonus produced by αC-conotoxin PrXA at 0.5 μg/g of mouse body weight is the only symptom preceding the rapid death of the animal. While two aptamers showed efficacy in reducing the myoclonus substantially, regardless of the mode of injection (concomitantly or differed), this observation may not necessarily remain valid for the toxin-induced mouse lethality. We therefore tested the effect of all four aptamers on mice lethality using the two paradigms of aptamer application (co-injection or differed by 1 min in tail vein injection). In an earlier report, we have illustrated that the D7 aptamer dose-dependently and efficiently prevented mice lethality induced by 0.5 μg toxin/g of mouse body weight [8]. The percentage of mice surviving the toxin injection was improved, as well as the delay between death occurrence and time of toxin injection for aptamer concentrations that did not fully prevent death. Herein, the same experiment was conducted with the D7 aptamer in the two conditions: (i) premixed with the toxin (simultaneous injection of the toxin and aptamer) and (ii) differed injection of the aptamer with regard to the toxin. 

Quite logically, the neutralizing capacity of the aptamer was improved in the premixed condition and the death occurrence time further delayed (Figure 4a,b). This stems from the fact that the aptamer does not have to cope with a differential organ distribution at the start of the toxin administration and with toxin recognition hindrance linked to a complex in vivo environment (plasma buffering for instance). As a result, we tested all other aptamers. When the toxin and aptamer B4 were injected simultaneously, low concentrations of B4 aptamer (less than 2 µg of aptamer/g of mouse) rescued up to 83.3% of mice from death (5/6) and concentrations equal or greater than 2 µg/g of aptamer B4 rescued all mice (Figure 4c, green dots). At low aptamer concentrations, B4 had the potency to delay toxin-induced death up to 78.0 ± 0.1 min (Figure 4d, green dots). In contrast, when aptamer B4 was injected separately (in mouse tail vein, 1 min after toxin injection), it no longer was able to rescue mice from death (0/6 mice, Figure 4c, red dots) and had only a slight potency to delay toxin-induced death up to 14.2 ± 0.6 min (Figure 4d, red dots). In spite of the fact that they did not show efficacy in reducing toxin-induced myoclonus at a concentration of 8 μg/g of body weight, we evaluated the ability of aptamers D3 and A5 to reduce or delay toxin-induced death. When both the toxin and the D3 aptamer were injected simultaneously, low concentrations of aptamer D3 (less than 2 µg of aptamer/g of mouse body weight) rescued 66.6% of mice from death (4/6 mice) and concentrations greater than 2 µg/g of aptamer D3 rescued all mice (Figure 4e, green dots). According to the fitted curve, the efficacy of this aptamer seemed lower than the ones observed for both D7 and B4, which may explain why it did not seem to protect from myoclonus. Alternatively, the dose of D3 used for myoclonus was maybe too low. For low D3 aptamer concentrations that did not prevent death at 100%, the aptamer delayed toxin-induced death up 72.7 ± 7.1 min (Figure 4f, green dots). When aptamer D3 was injected separately (in mouse tail vein, 1 min after toxin injection), it possessed no rescue ability: survival from toxin-induced death in 0/6 mice (Figure 4e, red dots) and had only a minor potency to delay toxin-induced death up to 12.8 ± 0.9 min (Figure 4f, red dots). In this respect, this aptamer behaved very much like B4. Contrary to the aptamers B4 and D3, the A5 aptamer was not able to rescue mice from death regardless of the mode of administration (mixture or separately, 0/6 mice for each mode of injection, Figure 4g). The route of administration of the aptamer did not affect his potency to delay toxin-induced death up to 12.2 ± 1.2 min for mixed injection or 11.9 ± 0.9 min for separate injection (Figure 4h). However, the delay in death occurrence observed with the A5 aptamer can be interpreted as a sign of small efficacy.

## 3. Discussion

Aptamers are unusual ligands as they are built of nucleotide sequences. The molecular basis for their peptide or protein recognition do not obey the same chemical rules than protein/protein interactions. The aptamers that have been investigated in this study arise from a CE-SELEX selection procedure against the synthetic αC-conotoxin PrXA and were classified according to potential secondary structures into three different groups [8]. While the structural basis of the interaction of these aptamers with the toxin are still unknown, one can surmise that the interaction is based mainly on electrostatic interactions since the peptide is mainly basic (5 positive charges *versus* 2 negative ones in the amino acid sequence) and the aptamer charged negatively. In addition, the interaction must be based on adopted secondary/tertiary structures that structurally recognizes a part of the disulfide-bridged-stabilized structure of the toxin otherwise all aptamers would be found interacting with the toxin, regardless of their sequence. What remains unclear however is whether these aptamers possess *de facto* the correct structure for αC-conotoxin PrXA recognition or whether the toxin forces them to adopt the proper fold that allows a high affinity interaction between the two ligands. Another issue that remains unknown is how the aptamers fold and behave in vivo, in a complex molecular and cellular environment, as they were not selected to interact with the toxin in such a complex mixture. One could expect problems regarding folding, bio-distribution, stability and toxin recognition (especially if the toxin itself is taken in charge by plasmatic proteins). In addition, another unknown issue is what happens to a preformed aptamer/toxin complex upon injection in vivo? There would be no guarantee that the interaction is preserved and that the toxin is not released rapidly for keeping its toxic activity in vivo. In our earlier investigation, we had studied mostly a single aptamer but had not investigated aptamers that belong to each structural group on a systematic basis. It was important to complete this first study as we wanted to understand whether all toxin-interacting aptamers could be efficient neutralizers of the toxic effects or if only a fraction of them were efficient. In the first case scenario, we would have expected that interaction *per se* is a self-standing and sufficient condition, regardless of the domain of interaction on the toxin, indicating that aptamers probably act by altering the normal biodistribution of the toxin or the pharmacophore of the toxin required for interaction with the muscle type nAChR. In the second case scenario, with only a few aptamers being efficient, then we would hypothesize that aptamers differentially neutralize the toxin pharmacophore and/or the biodistribution of the toxin in vivo. Alternatively, we could also envision that toxin recognition is lost in vivo or that the aptamers lose their recognition capacities to the toxin once injected in vivo. As the data illustrate, it is the second scenario that prevails. In many ways, it is logic because there was no reason that all the selected aptamers would cover the pharmacophore of the toxin. Only aptamers D7 and B4 seemed to bind, at least partially, to the pharmacophore as both were able to shift the IC_50_ of toxin-mediated inhibition of ACh-triggered currents in TE671 cells. The complete lack of effects of the D3 and A5 aptamers is an indication that, albeit binding onto the toxin [8], they do not interfere with the critical toxin residues that are involved in ACh receptor binding. Of course, a definitive conclusion that two of these aptamers bind onto the pharmacophore of the toxin will await a complete alanine scan investigation of the activity of αC-conotoxin PrXA completed with structural investigations of toxin/aptamer complexes in solution. For the pharmacological evaluation, we had to develop a specific assay that take advantage of an automated patch clamp system and high-throughput electrophysiological recordings. It is at the price of a large number of recordings that it turned possible to reliably measure compound affinities in a variety of conditions (ACh alone, toxin alone and toxin pre-incubated with various aptamers). As a consequence, the findings were highly reproducible and comparable. The important heterogeneity in ACh-mediated current amplitude in TE671 cells forced us to establish a new paradigm for the evaluation of αC-conotoxin PrXA effects on ACh currents. Each cell had therefore to be its own control for the effect of the toxin. Thanks to the reversible binding of the toxin onto its site and the reproducibility of ACh current amplitudes in each cell, we could measure first the ACh-mediated currents in αC-conotoxin PrXA-treated TE671 cells followed by the amplitude measurements of the ACh-mediated currents in the absence of the toxin or toxin/aptamer mixtures. This protocol reliably provided an estimate of the blocking potency of the toxin and the neutralizing capabilities of the aptamers. The only difference we could observe between a first and second application of ACh was the reduction of desensitization to ACh for the second application which may indicate that the first application of ACh somehow modified, through a signaling mechanisms, the biophysical properties of the muscle-type nAChR. Importantly, it should be mentioned that this assay is now valid for the screening of compounds active on the nicotinic receptor, which is the primary function of the Syncropatch 384 from Nanion. The complexity of the response of the aptamers in vivo was even greater than in vitro. In short, there is no simple mechanistic relationship between the results in vitro (presence or lack of toxin neutralization) and what should be expected to occur in vivo (lack or presence of neutralization). Aptamers that affected the pharmacophore of the toxin turned to be efficient in neutralizing the toxin-induced myoclonus as well as mice lethality but with some subtle differences among D7 and B4. While D7 was efficient in all conditions (premixed injection of toxin and aptamer, and differed injections), B4 lacked efficacy in neutralizing αC-conotoxin PrXA-induced lethality in the differed mode of injection, while curiously preventing the appearance of myoclonus in this mode as well. We have no clear explanation for this discrepancy except that maybe the B4 aptamer may interact with the toxin in organs that are not critical for survival (abdominal muscles) and not where it is required to ensure mice survival (diaphragm for instance). A complete investigation of how the B4 aptamer affects the biodistribution of αC-conotoxin PrXA in vivo would be required to investigate this issue. Surprisingly, a complete reverse observation was evident for the D3 aptamers. While this aptamer was not at all active in vitro, it was able to prevent mice death in vivo in the premixed mode of injection while not impacting the appearance of myoclonus. We therefore conclude that some toxin binding aptamers, while not being able to block the toxin pharmacophore, probably alter the normal biodistribution of the toxin in vivo and the regular organ access by the toxin. Here, the spectra of in vivo effects were thus opposite to that observed for the B4 aptamer, indicating that the biodistribution alterations should not be identical among aptamers. Notably, however, D3 had no effect at all in the differed mode of injection indicating that it either was unable to localize and bind the toxin in vivo, or that it was too late to alter its biodistribution. Finally, the most coherent aptamer was A5 as it had no effects at all both in vitro and in vivo.

## 4. Materials and Methods

### 4.1. Animals

Adult male and female *Mus musculus* Swiss mice weighing 20 to 23 g were used. They were kept in 12 h light/dark cycle. Each animal was used only once and was handled according to standard protocols for the use of laboratory animals. The investigation conforms to the Guide for the Care and Use of Laboratory Animal published by the US National Institutes of Health (NIH; publication No. 85-23, revised 1996) and received approval of the Cameroon National Ethical Committee (Yaoundé, Cameroon) for animal handling and experimental procedure (Ref No. FW-IRB00001954). All efforts were made to minimize animal suffering and reduce the number of animals used. At the end of each experiment, the general behavior of each mouse was observed for each condition, for 1, 4 and 24 h. Any adverse effects were noted as described previously [8].

### 4.2. Cell Culture

TE671 cells were maintained in Dulbecco’s Modified Eagle’s Medium (DMEM) supplemented with 10% fetal calf serum, 1 mM pyruvic acid, 4 mM glutamine, 10 IU/mL penicillin and 10 μg/mL streptomycin (Gibco, Grand Island, NY, USA), and incubated at 37 °C in a 5% CO_2_ atmosphere. For electrophysiological recordings, cells were detached with trypsin and floating single cells were diluted (~300 k cells/mL) in medium containing (in mM): 4 KCl, 140 NaCl, 5 Glucose, 10 HEPES (pH 7.4, osmolarity 290 mOsm).

### 4.3. Toxin and Aptamers Preparation

αC-conotoxin PrXA (Smartox Biotechnology, Saint Egrève, France) was dissolved in buffer A solution: 0.09% saline + 5 mM MgCl_2_. The volume of injection used was 10 μL/g mouse body weight (b.w.). For injection of toxin and aptamers (Eurofins Genomic, Ebersberg, Germany) in the same mixture, the toxin was incubated in buffer A with varying amounts of aptamers for 2 h at 4 °C (0.5 μg/g mouse b.w. for aptamers A5, B4, D3 and D7 and from 0.0625 to 8 μg/g mouse b.w.). For intravenous injection of aptamers, aptamers were prepared in buffer A.

### 4.4. Aptamer Potency to Neutralize Abdominal Myoclonus

To study the effects of aptamers neutralization potency on abdominal myoclonus, two methods of injection were tested. First, αC-conotoxin PrXA (0.5 µg/g of body weight) and each concentration (0.0625–0.125–0.25–0.5–1–2–4–8 µg/g of body weight) of aptamer (B4 or D7) were mixed and injected intraperitoneally in mice. For the second method of administration, αC-conotoxin PrXA was injected intraperitoneally and 1 min later aptamer was injected intravenously. Mice were split in ten groups of six mice (3 females and 3 males). Group I was treated with vehicle (10 µL/g i.p.) and group II received αC-conotoxin PrXA (0.5 μg/g mouse b.w.) and vehicle. Group III-X were treated with B4 or D7 aptamers. Immediately after the administration, each mouse was observed for 30 min and the number of abdominal myoclonus per animal was recorded. 

### 4.5. Aptamer Potency to Neutralize Toxin-Induced Lethality

Lethality and latency to death induced by αC-conotoxin PrXA were determined by intravenous injection of toxin at varying concentrations (0.005–0.008–0.01–0.05–0.1–0.5–1–1.5 μg/g of body weight). In order to examine aptamers potency to neutralize toxin induced-lethality, two methods of injection were tested as we did previously for abdominal myoclonus.

### 4.6. Automated Patch-Clamp Recordings

Whole–cell recordings were used to investigate effects of αC-conotoxin PrXA and aptamers on ACh-induced current. Automated patch-clamp recordings were performed using SyncroPatch 384PE (Nanion, München, Germany). Chips with single-hole high resistance (~10 MΩ) were used for TE671 cells recording. Pulse generation and data collection were performed with PatchControl384 v1.5.2 (Nanion) and Biomek v1.0 (Beckman Coulter, Brea, CA, USA). Whole-cell recordings were conducted according to the Nanion procedure. Cells were stored in a cell hotel reservoir at 10 °C with shaking speed at 60 RPM. After initiating the experiment, cell catching, sealing, whole-cell formation, liquid application, recording, and data acquisition were performed sequentially. αC-conotoxin PrXA and aptamers were purchased from Eurofins. The intracellular solution contained (in mM): 10 CsCl, 110 CsF, 10 NaCl, 1 MgCl_2_, 1 CaCl_2_, 10 EGTA and 10 HEPES (pH 7.2, osmolarity 280 mOsm), and extracellular solution contained (in mM): 140 NaCl, 4 KCl, 2 CaCl_2_, 1 MgCl_2_, 5 glucose and 10 HEPES (pH 7.4, osmolarity 298 mOsm). Whole-cell and patch experiments were performed at a holding potential of −60 mV at room temperature (18–22 °C). Currents were sampled at 10 kHz. Each molecule, acetylcholine (100 µM, Sigma, Saint-Quentin Fallavier, France), αC-conotoxin PrXA and aptamers, was prepared in extracellular solution supplemented in 0.3% bovine serum albumin in 384-well compound plate. Working compound solution was diluted 3 times in recording well by adding 30 μL to 90 μL external solution to reach final concentration. αC-conotoxin PrXA and aptamers were applied 15 min prior to ACh addition. ACh was applied as 5-s pulse and rapidly washed using stacked addition to avoid nAChR desensibilization. Data were analyzed with DataController384 V1.6.0_B9 and GraphPad. Dose-response relationships were fit to a sigmoidal equation: Y = Bottom + (Top − Bottom)/(1 + 10^((LogIC_50_ − X) × HillSlope)).

### 4.7. Statistical Analysis

Data are presented as means ± s.e.m. For comparison of multiple groups, One-way ANOVA was used followed by Tukey *post-hoc* test. A *P* value below 0.05 was considered significant.

## 5. Conclusions

All in all, this study was quite informative on the various profiles of aptamer neutralization capabilities one may expect for peptide toxins. A comparison with antibodies would be premature at this stage, but one may expect that venom sera probably contribute to toxin neutralization in a same way as aptamers. It is not necessary to have neutralizing agents that bind onto the pharmacophore although it should be considered an advantage of course. This study highlights future prospects in the investigation of how aptamers act on toxins. It seems quite important to gather structural data to see what are the molecular determinants of aptamer/peptide interaction. This study should be coupled to a better knowledge of the pharmacophore of the peptide. Finally, new tools should be developed to see how the toxin or aptamers distribute in the mouse body and how the aptamers affect the biodistribution of the toxin. Additional data on the pharmacokinetics and mode of elimination would be helpful informative data.

## Figures and Tables

**Figure 1 molecules-24-00229-f001:**
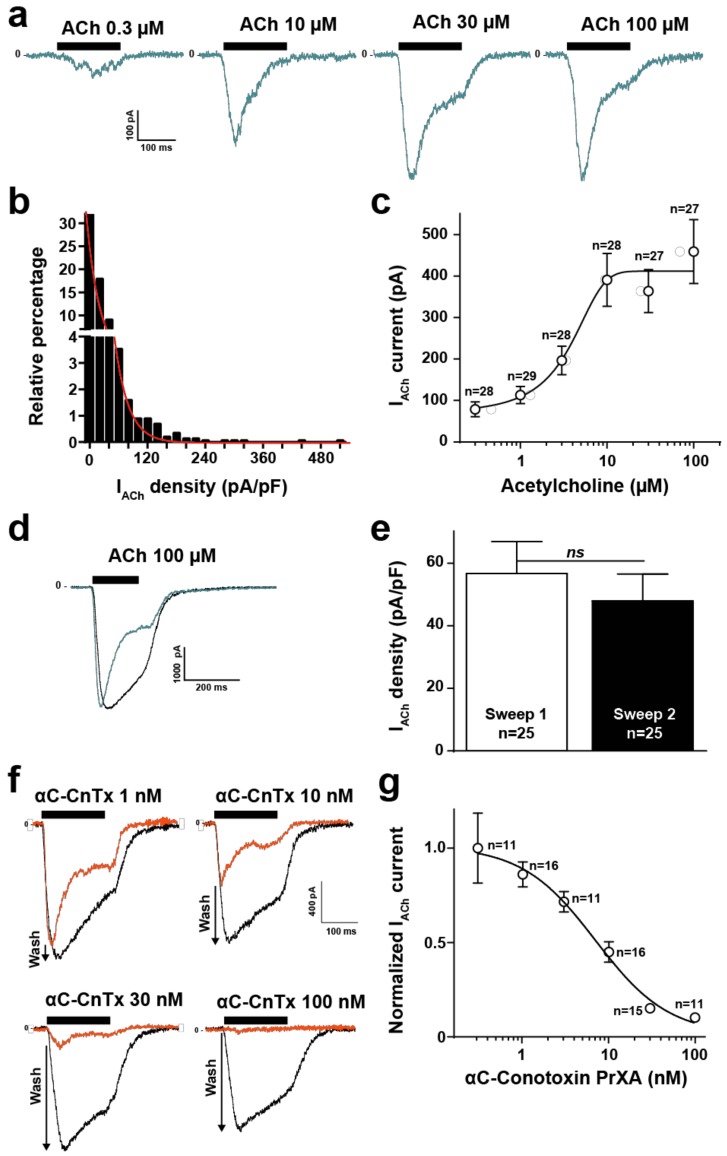
ACh and αC-conotoxin PrXA effects on TE671 cells. (**a**) Representative ACh-mediated currents at increasing ACh concentrations. (**b**) Histogram of current densities elicited by 100 μM ACh. (**c**) Dose-response of ACh-mediated inward currents. Mean ± SEM (*n* = 167 cells). (**d**) Representative inward currents mediated by two applications of 100 μM ACh (blue trace represents the second application). 10 min of rest time between the two applications and extensive renewal of extracellular solution after the first application. (**e**) Average current densities for first and second application of 100 μM ACh. (**f**) Representative currents traces for the first application of 100 μM ACh in the presence of increasing concentrations of αC-conotoxin PrXA (orange trace) and for the second application of ACh after extensive washout of αC-conotoxin PrXA (black trace). Arrows indicate the extent of current recovery, highest at highest toxin concentration. (**g**) Dose-response curve for the αC-conotoxin PrXA-mediated block of ACh response (*n* = 80 cells).

**Figure 2 molecules-24-00229-f002:**
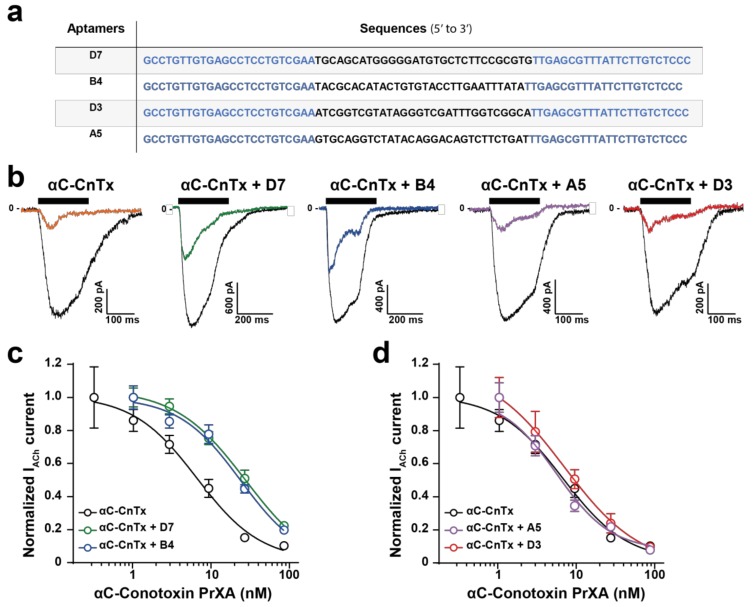
Impact of DNA aptamers on the αC-conotoxin PrXA-mediated inhibition of ACh response in TE671 cells. (**a**) Nucleotide sequences of the toxin-binding aptamers. Blue lettering corresponds to the conserved primers used in the CE-SELEX experiments, while normal lettering corresponds to the differentiating sequences between the aptamers. (**b**) Representative inward currents mediated by two applications of 100 μM ACh. The first application occurs after a 20-min pre-incubation of the cells in the presence of 30 nM αC-conotoxin PrXA and 300 nM aptamer (colored trace), while the second application occurs after thorough washout of both the toxin and the aptamer (black trace). (**c**) Dose-response curves of αC-conotoxin PrXA alone (black) or combined with aptamers (300 nM for each, D7: green, B4: blue) on I_ACh_ currents. For each concentration, *n* = 11–24. The data are fitted by a sigmoid curve and yield IC_50_ values of 6.9 ± 1.5 nM (toxin), 30.9 ± 2.2 nM (toxin + D7) and 26.5 ± 1.7 nM (toxin + B4). (**d**) Dose-response curves of αC-conotoxin PrXA alone (black) or combined with aptamers (300 nM for each, A5: pink, D3: orange) on I_ACh_ currents. For each concentration, *n* = 6–16. Fit of the data yield IC_50_ values of 5.8 ± 1.2 nM (toxin + A5) and 9.0 ± 1.5 nM (toxin + D3).

**Figure 3 molecules-24-00229-f003:**
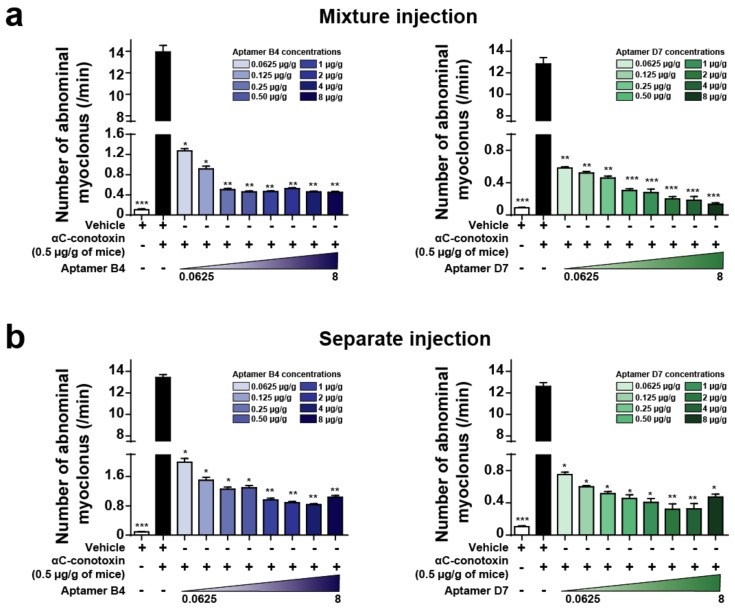
Aptamer neutralization of αC-conotoxin PrXA-induced myoclonus. (**a**) Number of abdominal myoclonus per min after αC-conotoxin PrXA injection alone or combined with increased concentrations of aptamers B4 (**left**, bleu) or D7 (**right**, green). *n* = 6 mice for each condition. (**b**) Number of abdominal myoclonus per min after αC-conotoxin PrXA injection alone or followed by increased concentrations of aptamers B4 (**left**, blue) or D7 (**right**, green). *n* = 6 mice for each condition. One-way ANOVA, * *p* < 0.05, ** *p* < 0.01, *** *p* < 0.001 *versus* αC-conotoxin PrXA injection alone (black bar).

**Figure 4 molecules-24-00229-f004:**
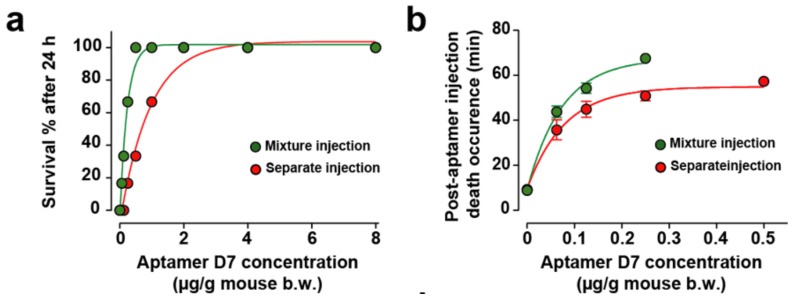
In vivo evaluation of aptamer neutralization properties. (**a**,**b**) Dose-response curve illustrating aptamer D7 neutralization properties on death rescue (**a**) and latency to death (**b**). One set of data was previously published in [8]. (**c**,**d**) Dose-response curve illustrating aptamer B4 neutralization properties on death rescue (**c**) and latency to death (**d**). (**e**,**f**) Dose-response curve illustrating aptamer D3 neutralization properties on death rescue (**e**) and latency to death (**f**). (**g**,**h**) Dose-response curve illustrating aptamer A5 neutralization properties on death rescue (**e**) and latency to death (**f**). Aptamers and toxin were administrated by intraperitoneal injection (mixture injection condition, green symbols and lines) or aptamers were injected by the intravenous route 1 min after intraperitoneal toxin injection (red symbols and lines). For each aptamer, αC-conotoxin PrXA concentration of 0.5 µg/g of mouse body weight was used. *n* = 6 mice for each aptamer concentration. *n* = 54 mice for each route of administration.

**Table 1 molecules-24-00229-t001:** Mice phenotypic modifications after intravenous injection of αC-conotoxin PrXA.

Treatments	Concentration (µg/g b.w.)	Sex	D/T	Mortality Latency (min)	Toxic Symptoms
Control	–	Male Female	0/3 0/3	– – – –	None
αC-conotoxin PrXA	0.005	Male Female	3/3 0/3	>23, <25	Hypoactivity, asthenia, tremors, loss of the righting reflex, myoclonus, exophthalmos
	0.008	Male Female	3/3 0/3	>23, <24	Hypoactivity, asthenia, tremors, loss of the righting reflex, myoclonus, exophthalmos
	0.01	Male Female	2/3 1/3	>10, <12 >12, <12	Hypoactivity, asthenia, tremors, loss of the righting reflex, myoclonus, exophthalmos
	0.05	Male Female	3/3 2/3	>1, <3 >1, <2	Hypoactivity, asthenia, tremors, loss of the righting reflex, myoclonus, exophthalmos
	0.1	Male Female	3/3 2/3	>1, <3 >1, <2	Hypoactivity, asthenia, tremors, loss of the righting reflex, myoclonus, exophthalmos, salivation and syncope
	0.5	Male Female	3/3 3/3	>1, <2 >1, <2	Hypoactivity, salivation, tachypnea, tremors, loss of the righting reflex, myoclonus, exophthalmos, salivation and syncope
	1	Male Female	3/33/3	>1, <2 >1, <2	Hypoactivity, salivation, tachypnea, tremors, loss of the righting reflex, myoclonus, exophthalmos, salivation and syncope
	1.5	Male Female	3/3 3/3	>1, <2 >1, <2	Salivation, tachypnea, tremors, loss of the righting reflex, myoclonus, exophthalmos, salivation and syncope

D/T = dead/treated mice; None = No toxic symptoms during the observation period; Mortality latency = time to death (in minute) after intravenous injection (10 µL/g) of αC-conotoxin PrXA to male and female mice. Control group received vehicle (intravenous injection, 10 µL/g).

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
