# Peer review of "Aptamer Efficacies for In Vitro and In Vivo Modulation of αC-Conotoxin PrXA Pharmacology"

_molecules, 2019, doi:10.3390/molecules24020229_

Round 1
Reviewer 1 Report
Manuscript is well written and presented. Results seem to be quite consistent in general. The subject is quite interesting within the area of venom antidotes.
I have however some suggestions and questions:
As far as this reviewer knows, the first demonstration that an aptamer can be an effective drug against a peptide toxin was published in 2000. The corresponding article (In vitro selection of RNA molecules that inhibit the activity of ricin A-chain. Hesselberth JR, Miller D, Robertus J, Ellington AD. J Biol Chem. 2000 275: 4937-42) should be cited and mentioned in the introduction (lines 80-85, for example).
Fig. S1a and b. Should be explained what is the meaning of i.v., i.p. and s.c. in the legend. The dotted green and red graphs, how were they obtained? What do they mean? The green dots in Fig. S1a, why do not show error bars?
I am afraid that some graphs must be missing from Fig. 4. Most especially those ones corresponding to mixture injections in panels a and b. I do not see either the dotted lines mentioned in the text (lines 301-310).
Author Response
Reviewer N°1
C1 - Manuscript is well written and presented. Results seem to be quite consistent in general. The subject is quite interesting within the area of venom antidotes. I have however some suggestions and questions: As far as this reviewer knows, the first demonstration that an aptamer can be an effective drug against a peptide toxin was published in 2000. The corresponding article (In vitro selection of RNA molecules that inhibit the activity of ricin A-chain. Hesselberth JR, Miller D, Robertus J, Ellington AD. J Biol Chem. 2000 275: 4937-42) should be cited and mentioned in the introduction (lines 80-85, for example).
R1 : We thank the reviewer for this suggestion. This reference has now been added in the introduction. The major difference however with this study remains that our data are the first proof of concept in vivo.
C2 - Fig. S1a and b. Should be explained what is the meaning of i.v., i.p. and s.c. in the legend. The dotted green and red graphs, how were they obtained? What do they mean? The green dots in Fig. S1a, why do not show error bars?
R2 : The meaning of i.v., i.p. and s.c. is now given in the legend. Dotted lines are fits through the data obtained by i.p. or s.c. injection modes of the toxin (these data are published earlier. We just show the fits). There are no error bars for the green dots because these are percentages. This is now all stated in supplementary figure 1 legend.
C3 - I am afraid that some graphs must be missing from Fig. 4. Most especially those ones corresponding to mixture injections in panels a and b. I do not see either the dotted lines mentioned in the text (lines 301-310).
R3 : The reviewer is correct. Same comment that reviewer 2. We modified the text in order to avoid this discrepancy and we added the plots on Fig. 4.
Reviewer 2 Report
Please improve the quality of the figures. The figures are at very poor resolution. In some figures it is impossible to read all of the text, making it hard to interpret your results.
Several references were made to the aptamers being separated into distinct classes by secondary structure, however no indication of the structural features that defined these classes is provided, nor are the secondary structures. It would be beneficial to provide this information, as well as to indicate if the structures are predicted based on computational folding algorithms alone or supported by biochemical probing methods.
In the introduction (lines 82-83) you state you are working with the toxin aC-conotoxin PrXA. It would benefit the paper to explain the origin, mode of action, and significance of particular this toxin, and would add to the motivation for this work.
Your experimental setup for measuring current changes in response to ACh addition via patch-clamp recordings involves first adding the toxin and performing a measurement, and then washing away the toxin and performing a second measurement as a control. Is there a rational for not taking the control measurement first, before any toxin is added, to insure that no residual toxin affects the control measurement?
In line 284 the concentration for D3 and A5 is given in units of uM, while all other preceding doses are given in units of ug/g. It would be easier to compare if all concentrations were given in ug/g.
Figure 1f appears to be misrepresented in the figure caption. It states that the “black curve” is the first measurement after addition of ACh and various concentrations of toxin. However, it is my understanding that the orange curve is actually the first measurement, as the caption further states that the arrow indicates recovery from the orange trace to the black trace. Please clarify and ensure that both the text and figure accurately depict what was done.
Lines 306-307 state that the experiment was conducted with the D7 aptamer premixed with toxin. However, the corresponding figure, Figure 4a,b, states that the data is representative of separately injected aptamer and toxin. Please clarify.
Lines 308-309 state that the neutralizing capacity of the aptamer was improved and the death occurrence time was further delayed, and references figures 4a,b. However, there is only a single data set presented in these figures. There is nothing to compare it to in order to illustrate its improvement. This dataset for comparison should be added. If this is in reference to the results of the previous publication, it may be beneficial to plot this data for comparison purposes and indicate that it was previously published.
Several places with grammatical issues that need to be addressed.
Author Response
Reviewer N°2
C1 - Please improve the quality of the figures. The figures are at very poor resolution. In some figures it is impossible to read all of the text, making it hard to interpret your results.
R1 : We have improved the quality of these figures. The first set was meant to be light for submission and reviewing. Thank you for noticing this point.
C2 - Several references were made to the aptamers being separated into distinct classes by secondary structure, however no indication of the structural features that defined these classes is provided, nor are the secondary structures. It would be beneficial to provide this information, as well as to indicate if the structures are predicted based on computational folding algorithms alone or supported by biochemical probing methods.
R2 : All of the aptamers isolated so far display stem-loop (see for example : Gopinath SC, Awazu K, Fujimaki M. Sensors, 12, 2136-51, 2012) or G-quadruplex (see for example: Tucker WO, Shum KT, Tanner JA. Curr. Pharm. Des. 18, 2014-26, 2012) structures. These folding motifs are commonly identified by structural and biochemical methods (see for example: D.J. Patel, Curr. Opin. Chem. Biol. 1, 32-46, 1997) or estimated by computational tools (Mfold for example: see Zuker M. Nucleic Acids Res. 31, 3406-3415, 2003). In our work, the aptamer structures were predicted by using the Mfold web server. All four of our aptamers display stem loops. The belong to different families according the nucleotide sequence. These informations have been added in the manuscript and some corrections introduced to be more factual.
C3 - In the introduction (lines 82-83) you state you are working with the toxin aC-conotoxin PrXA. It would benefit the paper to explain the origin, mode of action, and significance of particular this toxin, and would add to the motivation for this work.
R3 : The reviewer is right. Although the toxin was introduced in a previous publication, it is indeed good to summarize some of its most interesting properties. This is now done in the introduction at the suggested place.
C4 - Your experimental setup for measuring current changes in response to ACh addition via patch-clamp recordings involves first adding the toxin and performing a measurement, and then washing away the toxin and performing a second measurement as a control. Is there a rational for not taking the control measurement first, before any toxin is added, to insure that no residual toxin affects the control measurement?
R4 : Initially, we were concerned that a second application would lead to a smaller response. If this would occur, in addition to the reduction in current amplitude triggered by the toxin, it would be difficult to assess the real contribution of the toxin. The main issue with a first application of ACh in the absence of the toxin is that ACh also needs to be extensively washed to avoid long-term desensitization. Finally, the hill coefficient we obtain for the block by the toxin, close to 1, illustrates that there should be no residual toxin left for the second application of ACh (now added in the manuscript). Also, the average current amplitudes obtained after washout of the highest concentration of the toxin is very close to the average current amplitude obtained after the second application of ACh in the presence of the lowest concentration of toxin. These two data have been included in the manuscript to illustrate that the protocol is reliable and that there is no residual toxin left in the chamber and that full reversibility is obtained. We thank the reviewer for coming up with this pertinent comment that improves the value of our findings.
C5 - In line 284 the concentration for D3 and A5 is given in units of uM, while all other preceding doses are given in units of ug/g. It would be easier to compare if all concentrations were given in ug/g.
R5 : This was a mistake in our initial version. It has now been corrected. We thank the reviewer for noticing this discrepancy.
C6 - Figure 1f appears to be misrepresented in the figure caption. It states that the “black curve” is the first measurement after addition of ACh and various concentrations of toxin. However, it is my understanding that the orange curve is actually the first measurement, as the caption further states that the arrow indicates recovery from the orange trace to the black trace. Please clarify and ensure that both the text and figure accurately depict what was done.
R6 : The reviewer is perfectly right. Again thank you for noticing this point. It has been corrected accordingly.
C7 - Lines 306-307 state that the experiment was conducted with the D7 aptamer premixed with toxin. However, the corresponding figure, Figure 4a,b, states that the data is representative of separately injected aptamer and toxin. Please clarify.
R7 : See Response to C8 also. We have added the missing data and corrected the text accordingly.
C8 - Lines 308-309 state that the neutralizing capacity of the aptamer was improved and the death occurrence time was further delayed, and references figures 4a,b. However, there is only a single data set presented in these figures. There is nothing to compare it to in order to illustrate its improvement. This dataset for comparison should be added. If this is in reference to the results of the previous publication, it may be beneficial to plot this data for comparison purposes and indicate that it was previously published.
R8 : We have added the data set.
C9 - Several places with grammatical issues that need to be addressed.
R9 : We did our best to improve the grammar.